# DiffDeID: a Multi-conditional Diffusion-based Method for High Fidelity Face De-identification with Diversity

## Abstract

Face de-identification is a critical task that aims to obscure true identities while preserving other facial attributes. Current methodologies typically involve disentangling identity features within a latent space and leveraging adversarial training to balance privacy with utility, often at the cost of a trade-off between two. To surmount these limitations, we introduce DiffDeID, a novel approach grounded in diffusion models. This method incrementally safeguards identity and sustains utility, all while ensuring enhanced interpretability. Our method employs a Latent Diffusion-based ID Sample to generate authentic identity embeddings that are obfuscated from the original identity, thereby providing users with diverse options. Additionally, a multi-condition diffusion model is utilized for facial images, ensuring the retention of image utility. We further introduce a novel training and inference paradigm, utilizing the unified architecture tailored for video facial de-identification tasks. The robustness of our method is attributed to its powerful 3D prior and meticulous generation design, enabling natural identity protection, generation of high-quality details, and robustness across various attributes. Through extensive experimentation, we demonstrate that DiffDeID surpasses previous methodologies.

## 1 Introduction

As computer vision and image understanding technologies evolve, the risk of unauthorized disclosure of personal information through images has significantly increased. The use of facial data is highly valuable in a multitude of applications, such as emotion recognition and general face detection, but it also poses challenges related to privacy protection. Face de-identification, or facial anonymization, addresses this issue by removing identifiable details from facial images while preserving other critical aspects such as the context, actions, and non-identifying features like pose and expressionAgrawal & Narayanan (2011). There are various methods to create de-identified faces that resemble real humans and maintain visual fidelity, compatible with different tasks and scenarios. As the earliest and most straightforward approaches, blurring smooths out the facial features across the imageHarmon & Julesz (1973); Frome et al. (2009), while pixelation involves enlarging the pixels to a size that obscures detailZhou & Pun (2020). These techniques are easy to implement but can significantly reduce the image quality and utility for other tasks.

Deep learning has emerged as a transformative approach in face de-identification, offering sophisticated solutions that significantly enhance the balance between privacy protection and data utility. The most well-known methods include encoder-decoders and generative adversarial networks (GANs). The encoder-decoders learn to compress and reconstruct images in a way that can be tailored to remove identity information while retaining other relevant facial featuresQiu et al. (2022). GANs are generally composed of the generator and discriminator, which attempt to achieve a balance between identity privacy protection and image authenticity through adversarial learningHukkelås et al. (2019); Wen et al. (2022b). However, the unstable minimax objective adversarial training process of GANs further exacerbates unrealistic textures of the results. The effectiveness of these methods heavily depends on the degree of disentanglement between identity and attributes, which is often unsatisfactory. Additionally, existing methods struggle to preserve the pose and expression of the source face images or videos.

To address the aforementioned challenges, we adopt a multi-conditional diffusion-based de-identification model (DiffDeID) to avoid the unstable training of GANs. We propose Latent Diffusion-based ID Sampler (LDS) to generate anonymous identity embedding, which produces diverse and realistic anonymous identity embeddings different from the source identity ones. Furthermore, we use the 3DMMs interpretable and decoupled parameter space, combined with the source face texture correlation coefficient and identity coefficient to construct 3D descriptor. We take advantage of the explainability and disentanglement of 3DMMs parameter space and combine texture correlation coefficient of source face and generated identity coefficient to construct 3D descriptors. To further supplement the texture and background information from the source face to the rendered face, we introduce a cross-attention mechanism to accurately match the correspondences between the source face and the rendered face. In summary, the main contributions of our work are described as follows:

- We propose a Latent Diffusion-based ID Sampler based on DDIM, capable of sampling multidimensional identity embeddings with diversity and realism, which serve as anonymized identity embeddings for subsequent replacement.

- We propose a DiffDeID pipeline based on a multi-conditional diffusion model for face de-identification, which is able to preserving the pose and expression information of the source face image while replacing the identity information, thereby achieving high-quality face image generation.

- We propose a new training and inference paradigm based on DiffDeID architecture to achieve face de-identification in video, ensuring a high-degree preserving of face attribute information.

- Sufficient experiments have proven the effectiveness of our method. Specifically, our approach can achieve face de-identification while highly preserving the pose and expression , outperforming state-of-the-art methods on multiple metrics.

## 2 RELATED WORKS

### 2.1 FACE DE-IDENTIFICATION

The earliest approaches to privacy protection in image data relied primarily on blurring and pixelation techniques to obscure sensitive informationGross et al. (2006); Newton et al. (2005). While these methods were straightforward and computationally efficient, they often fell short in terms of securityOh et al. (2016). The de-identification effects achieved by these methods were usually unnatural, detracting from the aesthetic quality or the practical utility of the images.

With the rapid advancement of deep learning, the evolution of neural network architectures and the burgeoning research in facial de-identification have paved the way for innovative approaches. Autoencoders can be used to obscure certain facial features by function by manipulating the compressed representationLiu et al. (2023); Nousi et al. (2020). Gafni et al. (2019) introduced a new face identity transformer for privacy protection that uses cryptographic embedding and multi-task learning based on autoencoders. In recent years, GAN-basedGoodfellow et al. (2020) face de-identification methodsWu et al. (2019); Maximov et al. (2020) have gained significant popularity. Cao et al.Cao et al. (2021) propose a personalized and invertible face de-identification method by manipulating the distangled identity embeddings and the following work IdentityMaskWen et al. (2022a) decouples identity and attribute information, using latent space operations based on passwords and privacy level parameters to encrypt or recover identity, followed by image reconstruction. Despite the promising advancements in GAN-based face de-identification methods, there remains a lack of clarity regarding the interpretability of identities and other facial representations in the latent space. Additionally, the limitations of the generator and the unstable nature of GAN training necessitate the design of specific intermediate representations and generators for each driving mode.

### 2.2 3D FACE RECONSTRUCTION

Monocular 3D face reconstruction entails the construction of a three-dimensional facial model from a two-dimensional image, where 3D Morphable Models (3DMMs) Blanz & Vetter (2023) hold a

dominant position. Additionally, there are approaches advocating for direct model-free reconstruction Sela et al. (2017) or those based on other innovative models Li et al. (2017). Compared to other structural representations, such as landmarks and segmentation maps, 3DMMs offer an interpretable and disentangled parameter space, enabling direct manipulation of facial features for specific tasks Lattas et al. (2021); Wang et al. (2022). Rendered facial images from 3DMMs provide richer semantic and explicit geometric details than vectorized parameters, reducing the complexity of model training.

## 2.3 Denoising Diffusion Models

Diffusion models Ho et al. (2020); Nichol & Dhariwal (2021) are popular generative models in synthesizing high-quality images, which operate on a generative probabilistic framework consisting of two main steps. Initially, data is progressively corrupted by incrementally adding small amounts of Gaussian noise over a series of time steps. Subsequently, a learning algorithm is trained to recover the data by gradually removing the noise over another series of time steps. Compared to GANs, diffusion models offer a more reliable and robust training process due to the absence of discriminators. They also avoid common issues associated with GANs, such as mode collapse or vanishing gradients. Following the significant success in unconditional image generation, diffusion models have been adapted to support conditional generation. Ho et al. Ho & Salimans (2022) further advanced these models by developing classifier-free guidance methods, enabling conditional editing without the need for a pre-trained classifier. Despite these advantages, diffusion models are slower in sampling due to the requirement of thousands of denoising steps in pixel space for a single sample. To address this, Song et al. Song et al. (2020) proposed DDIM to reduce sampling time and Rombach et al. Rombach et al. (2022) introduced Latent Diffusion Models (LDMs), which shift the training and inference processes to a compressed low-dimensional latent space to enhance computational efficiency. Diffusion models have demonstrated the ability to generate high-quality data across a wide range of data distributions. In this paper, we construct a face de-identification framework leveraging diffusion models, which is applied to anonymize face images and extended to the generation of de-identified videos.

## 3 Methodology

### 3.1 Overview

Face de-identification is a process designed to alter face images to protect individual identities. By generating a realistic yet identity-concealed version $I$ of the original image $I'$, it aims to maintain visual similarity while preventing the identity information of faces from being recognized by certain recognition tools. Fig1 illustrates the overall pipeline of the proposed paradigm DiffDeID, which can protect face privacy information with high fidelity. To effectively accomplish face de-identification, we meticulously integrate multidimensional conditions to ensure the protection of individual identities while navigating the nuances of privacy preservation.

### 3.2 Identity Protection Conditions

To make the identity information of the original image invisible to the model, we employ sophisticated masking techniques that artfully erase the facial elements associated with inherent individuality. Additionally, we incorporate a intermediate structural representation through the employment of face rendered from 3DMM to couple the de-identified identity with the attributes of the original face, guiding the model generation process of de-identified faces. To achieve a high degree of differentiation from the original identity while preserving the realistic essence of face appearance, we employ a Latent Diffusion-based ID Sampler, which is able to sample anonymized identities from the distribution of real human identities.

**Identity Masked Image.** Elements such as hairstyles, accessories and backgrounds, having a minimal association with personal identity, can encompass a considerable area of the visual space. This expansive presence markedly shapes our perception of visual similarity and profoundly impacts its further application.

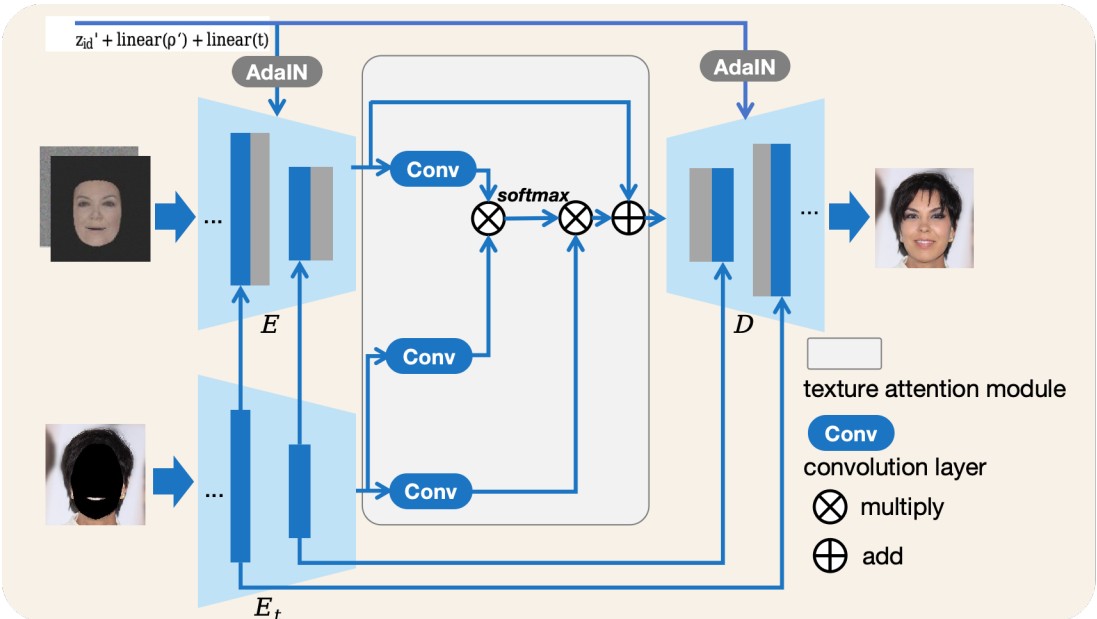

Figure 1: The architecture of DiffDeID. Given multimodal conditions, we initially procure the de-identified 3DMM coefficients $\rho'$ as described in section 3.2 and then project them into the rendered faces $I'_{3d}$, serving as an intermediate structural representation. To achieve realistic facial depictions, we develop a multi-conditional diffusion model, which learns geometric priors from the input through connected embeddings, conveys texture information from the source face $I$ via cross-attention mechanisms and supplements identity and geometric information using AdaIN.

We employ a facial mask predictor $\mathcal{M}_{mask}$ to obscure the facial region in the original image, effectively eliminating facial identity details, resulting in the identity mashed image $I_m$. Simultaneously, it preserves the image details excluding facial identity features to the greatest extent, such as hair and background.

Significantly, the area encompassing the mouth is preserved within the background context, facilitating a heightened realism in the reconstruction of the mouth region in face images. This is attributed to the model's ability to assimilate information from the maintained mouth region.

**3D Monocular Face Representation.** Recent methods optimize neural networks to extract three-dimensional parameters from facial images, thereby estimating three-dimensional facial descriptors for two-dimensional images. Following the work that utilized ResNet50 in D3DFR, we use it as the backbone to predict 3D Morphable Model (3DMM) coefficients. These include identity $\alpha \in \mathbb{R}^{80}$, expression $\beta \in \mathbb{R}^{64}$, texture $\delta \in \mathbb{R}^{80}$, illumination $\gamma \in \mathbb{R}^{27}$, and pose $p \in \mathbb{R}^{6}$. Notably, the original 3DMM lacked control over gaze direction. We explicitly model the gaze, providing a normalized directional vector from the eye center to the pupil in four dimensions $\omega \in \mathbb{R}^{4}$. Therefore, given the face image to be de-identified $I$, the output 3DMM coefficients is $\rho \in \mathbb{R}^{261}$:

$$\rho = \mathcal{M}_{3D}(I) = \{\alpha, \beta, \delta, \gamma, p, \omega\}, \tag{1}$$

where $\mathcal{M}_{3D}$ represents the 3DMM model. To achieve de-identification, we employ the Latent Diffusion-based ID Sampler discussed below to generate de-identified identity coefficients $\alpha'$, which is then used to replace the corresponding identity coefficients $\alpha$, yielding the de-identified 3DMM coefficients $\rho'$.

With initialized 3DMM coefficients, the 3D face shape $S$ and albedo texture $T$ could be parameterized as:

$$\mathbf{S} = \bar{\mathbf{S}} + \mathbf{P}_{id}\boldsymbol{\alpha'} + \mathbf{P}_{exp}\boldsymbol{\beta},$$
$$\mathbf{T} = \bar{\mathbf{T}} + \mathbf{P}_{t}\boldsymbol{\delta},$$

where $\bar{\mathbf{S}}$ and $\bar{\mathbf{T}}$ and denote the mean face shape and albedo texture. $\mathbf{P}_{id}, \mathbf{P}_{exp}$ and $\mathbf{P}_t$ are bases of identity, expression, and texture obtained from PCA.

The reconstructed 3D face model is then projected onto a 2D plane based on illumination $\gamma$ and pose $p$, resulting in the rendered face image $I'_{3d}$. Given the capacity of 3DMM to offer a parameter space that is both explainable and disentangled, we are enabled to meticulously preserve features that are agnostic to individual identity.

**Latent Diffusion-based ID Sampler.** Unlike previous industry work that proposed mapping-based obfuscation methods, we propose a DDIM-based generative model to construct diverse de-identified identity embeddings that differ from the original vectors and exhibit real human characteristics.

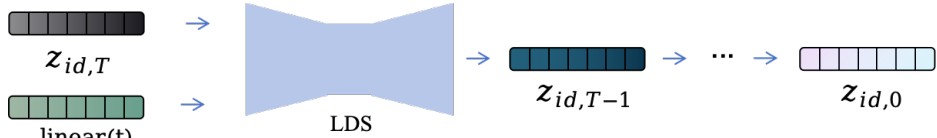

Figure 2: Overview of out Latent Diffusion-based ID Sampler (LDS).

The Latent Diffusion-based ID Sampler(LDS) samples identities from a normal distribution with zero mean and unit variance. LDS constructs identity feature latent vectors in the latent space and is trained using the following loss function:

$$L_{\text{LDS}} = \sum_{t=1}^{T} \mathbb{E}_{\mathbf{z}_{id}, \epsilon_t}[\|\epsilon_d(\mathbf{z}_{id,t}, t) - \epsilon_t\|_1], \tag{2}$$

where $t$ represents the current step of LDS, $\mathbf{z}_{id}$ represents the identity latent vector, $\mathbf{z}_{id,t}$ represents the latent vector representation at the $t$-th step, $\epsilon_t$ represents the random noise added at the $t$-th step of the diffusion process, and $\epsilon_t$ represents the noise predicted by LDS at the $t$-th step.

Similarly, the identity representation of 3DMM coefficients is modeled in the latent space and trained with the following loss function:

$$L_{\text{LDS}} = \sum_{t=1}^{T} \mathbb{E}_{\alpha, \epsilon_t}[\|\epsilon_d(\alpha_t, t) - \epsilon_t\|_1]. \tag{3}$$

In our refined methodology, we initially standardize the latent distribution from LDS to have zero mean and unit variance. The de-identified identity embedding $z'_{id}$ and de-identifaied 3DMM identity representation $\alpha'$ are generated by sampling from LDS and unnormalizing them. Replacing the identity representation in 3DMM by $\alpha'$ results in the de-identifaied 3DMM coefficients:

$$\rho' = \{\alpha', \beta, \delta, \gamma, p, \omega\}. \tag{4}$$

Diverse identity embeddings can be obtained through LDS sampling, which can be filtered by posterior identity cosine similarity to ensure the de-identified identity embeddings are significantly divergent from the original identity. By extracting samples from the latent space which encapsulates the variance of genuine facial identities, the derived identity embeddings are endowed with an enhanced degree of fidelity.

### 3.3 DIFFUSION-BASED DE-IDENTIFICATION MODEL

Recent GAN-based methods tend to produce noticeable artifacts and degradation issues when reproducing complex textures and identity appearances under various extreme conditions like exaggerated pose and degradation problems. Therefore, we adopt a face de-identification model based on diffusion models. DiffDeID uses U-Net as the network backbone, incorporates texture attention blocks with a cross-attention mechanism, and applies adaptive instance normalization to inject identity and attribute features. In this section, we will describe the network structure of DiffDeID and the training details of the denoising process.

**Architecture.** Our multi-conditional diffusion-based model $\epsilon_\theta$ is based on U-Net, consisting of an encoder $E$ and a decoder $D$. The background texture encoder $E_t$ encodes the masked face image $I_m$ to obtain background features, which shares the same structure as the U-Net encoder $E$. Also, we introduce a texture attention module to accurately retain background texture information and mitigate the impact of feature misalignment. The input to this module includes noise features $F_n$ obtained from the U-Net encoder $E$ and background texture features $F_b$ obtained from $E_t$. Through three convolutional layers, the query $Q_n$ is extracted from $F_n$, while the key $K_b$ and value $V_b$ are extracted from $F_b$. $Q_n$ and $K_b$ are multiplied and passed through a Softmax layer to compute the correlation matrix, which is then multiplied by $V_b$ to obtain weighted features $F_{b\rightarrow n}$. $F_{b\rightarrow n}$ is then multiplied by a learnable parameter $\tau$ initialized to zero and added to $F_b$, resulting in the final feature representation $\hat{F}_n$:

$$F_{b\rightarrow n} = \text{Softmax}(Q_n K_b^{\text{T}})V_b = MV_b \tag{5}$$

$$\hat{F}_n = \tau F_{b\rightarrow n} + F_b \tag{6}$$

In addition, the de-identified identity embedding $z'_{id}$ and the de-identifaied 3DMM coefficients $\rho'$ augments the obfuscation of identity and incorporate supplementary information that refines the geometric details of the original face, specifically the gaze direction coefficient $\omega$. Incorporating the time embedding, we derive the condition $C = z'_{\text{id}} + \text{linear}(p') + \text{linear}(t)$, which is injected into U-net through the adaptive instance normalization (AdaIN):

$$\text{AdaIN}(F_n, C) = \sigma_c(C)\frac{F_n - \mu(F_n)}{\sigma(F_n)} + \mu_c(C), \tag{7}$$

where $\sigma(F_n)$ and $\mu(F_n)$ are the mean and variance of the input features, and $\sigma_c(C)$ and $\mu_c(C)$ are used to estimate the adapted mean and variance based on $C$. At this juncture, our model $\epsilon_\theta(Z_t, I_m, I_{3d}, p, z_{id}, t)$ adeptly integrates a constellation of conditions—comprising de-identified identity embeddings, the rendered face and the other attributes—to preidict the noise, thereby culminating in the generation of de-identified face images.

**Loss Function.** We first employ a conventional denoising loss, allowing the model to learn to transform the current noisy sample into less noisy sample at certain time step $t$:

$$\mathcal{L}_{\text{d}} = \|\epsilon_{\text{t}} - \epsilon_\theta(Z_t, I_m, I_{3d}, p, z_{id}, t)\| \tag{8}$$

where is the applied noise at step $t$. Besides, we define $\hat{\mathbb{Z}}_0$ as the final denoised face, thereby allowing further constraints on the model in the image dimension. The reconstruction loss is defined as the pixel-level $\mathcal{L}_2$ distance, ensuring that the generated image closely resembles the original image.

$$\mathcal{L}_{\text{rec}} = \|\hat{\mathbb{Z}}_0 - I\|_2. \tag{9}$$

Additionally, we define the perceptual loss as the $\mathcal{L}_2$ distance of the perceptual features:

$$\mathcal{L}_{\text{per}} = \|\Phi_{\text{vgg}}\hat{\mathbb{Z}}_0 - \phi_{\text{vgg}}(I)\|_2 \tag{10}$$

where $\Phi_{\text{vgg}}(\cdot)$ represents the pre-trained VGG16 Simonyan & Zisserman (2014) network. The overall model objective is a combination of the above losses:

$$\mathcal{L} = \lambda_d\mathcal{L}_{\text{d}} + \lambda_{rec}\mathcal{L}_{\text{rec}} + \lambda_{per}\mathcal{L}_{\text{per}}, \tag{11}$$

The hyper-parameters are set as follows: $\lambda_d = 10, \lambda_{rec} = 1, \lambda_{per} = 1$.

## 3.4 A NOVEL PARADIGM FOR REENACTMENT BASED ON DIFFDEID

In the realm of video processing, the conventional frame-by-frame face de-identified technique, which treats each video frame as a discrete entity, often leads to a disjointed and unconvincing visual output. To address this limitation, our research introduces a novel training and inference methodology applied to the DiffDeID framework for face reenactment, which enables a more nuanced and lifelike transformation of facial expressions, poses, and other dynamic attributes within the video sequence. It also ensures that the source identity embedding remains constant throughout the video stream processing.

First, we undertake the de-identified face image of the inaugural frame (denoted as $I_d$) and employ it as the input for the texture encoder to ensure consistent and coherent facial textures throughout the video sequence. Second, because we prioritize the dynamic identity-unrelated attributes of facial expressions when 3DMM coefficients, we integrate the expression $\beta_f$, pose $p_f$, and gaze direction $\omega_f$ extracted from the video frame $I_f$ with the identity $\alpha_d$, texture $\delta_d$ and illumination derived $\gamma_d$ from the de-identified image $I_d$ to construct the 3D face descriptors $\hat{\rho} = \{\alpha_{\mathfrak{d}}, \beta_{\mathfrak{f}}, \delta_{\mathfrak{d}}, \gamma_{\mathfrak{d}}, p_f, \omega_{\mathfrak{f}}\}$. Since we focus more on the dynamic changes of identity-independent attributes, the last condition $C$ does not include identity embedding, which can be expressed as $C = linear(\hat{\rho}) + linear(t)$. During the training phase, the loss function consists of the denoising loss, reconstruction loss and perceptual loss with the same setting as Eq. 11. During the inference process, we render the face from the 3D descriptors $\hat{\rho}$ to obtain $I_{3d}$ as the input to the U-Net encoder, take the de-identified face $I_d$ as the input to the texture encoder and inject the last condition $C$. Moreover, we utilize the parameters of adjacent frames as descriptors for the central frame like Ren et al. (2021), thereby smoothing the motion trajectory. Subsequently, utilizing the denoising steps, we adeptly transfer the attributes from the video stream onto the de-identified face, thereby achieving effective video de-identification.

## 4 EXPERIMENT

### 4.1 EXPERIMENTAL SETUP

**Datasets.** We leverage the high-quality CelebAMask-HQ datasetLee et al. (2020), which comprises 30,000 images with fine-grained mask annotations, to train our DiffDeID model for de-identification. For the purpose of testing, we utilize the FaceForensics++ datasetRossler et al. (2019), an extensive forensic dataset encompassing 1000 videos. This selection of datasets provides a rigorous benchmark for evaluating the performance and robustness of our face anonymization and reenactment methodologies. We employe the VoxCeleb1 datasetNagrani et al. (2017) for video de-identification, which encompasses over 20,000 videos. Within this extensive collection, we have specifically selected videos of high resolution, namely 720P. Following the preprocessing methodology outlined in FOMMSiarohin et al. (2019), these videos are cropped and resized to a uniform dimension of 256 × 256 pixels. This meticulous preparation yields a substantial dataset comprising 17,927 training videos and 491 test videos, which serve as a robust foundation for our experiments and evaluations in video de-identification task.

**Evaluation Metrics.** For face de-identification, we employ the Frechet Inception Distance (FID)Heusel et al. (2017) to evaluate the realism of the generated faces. Besides, we utilize the EXP, POSE, and GAZE to calculate the average Euclidean distance between the corresponding coefficients of the generated face and the original face. These metrics are instrumental in assessing the fidelity of the facial reenactment process. To measure the identity cosine similarity, we employ identity embeddings ID extracted using ArcfaceDeng et al. (2019). For video de-identification, we also adopt the cosine distance of id consistency, expression, pose, gaze and de-idenditfed id for evaluation. ID consistency distance is determined by computing the cosine distance between the identity embedding of the current frame in the de-identified video and the identity embedding of the first de-identified frame. The de-identification ID distance is quantified by calculating the cosine distance between the identity embeddings of the frames from the source video and those from the de-identified video. Expression, pose, gaze distance is evaluated by measuring the cosine distance between the expression, pose, gaze embeddings of the frames in the source video and the corresponding frames in the de-identified video.

**Implementation Details.** In the context of the DiffDeID diffusion model, the length of the denoising step $T$ is set to 1000, with both training and inference processes employing a linear noise schedule. It is noteworthy that the denoising loss is utilized exclusively during the initial phase of training to stabilize the training process. The transition to incorporating the reconstruction loss and the perceptual loss is initiated only when the denoising noise drops below 0.05. Additionally, the UNet of DiffDeID is designed to process images at a resolution of 256×256 pixels with a downsampling ratio of 16. We employ the Adam optimizerKingma & Ba (2014) to train our model for a total of 200,000 iterations, with the learning rate configured at $2 \times 10^{-4}$ and the batch size configured at

8. For the DiffDeID model tailored for video de-identification, we adopt the aforementioned settings and proceed by randomly sampling faces from the VoxCeleb1 video dataset for training.

## 4.2 DE-IDENTIFICATION ANALYSIS

In the context of LDS model, identity embeddings are sampled from the real human face identity space through being generated from pure Gaussian noise. For each test face image, distinct identity embeddings are utilized to produce ten anonymized facial images with increasing de-identified levels. Various statistical metrics are then computed at each level to evaluate the effectiveness and progression of the de-identification process.

The qualitative findings are shown in Fig 3. As the de-identified level increases, the geometric differences between the de-identified and original faces become more noticeable, while non-identity attributes like hairstyle and background remain unchanged. The de-identified images consistently maintain high quality, nearly matching that of the original images. The de-identified effects are notably diverse, encompassing a range of variations such as changes in beard, skin tone or eye size.

The quantitative results are presented in Fig 4. It can be seen that the degree of identity protection varies with the de-identified level. Notably, the attributes unrelated to identity, such as expression and pose, is consistently maintained at a excellent level. As the de-identified level increases, there is a slight upward trend in the distance of utility attributes, but the impact is minimal and remains good throughout.

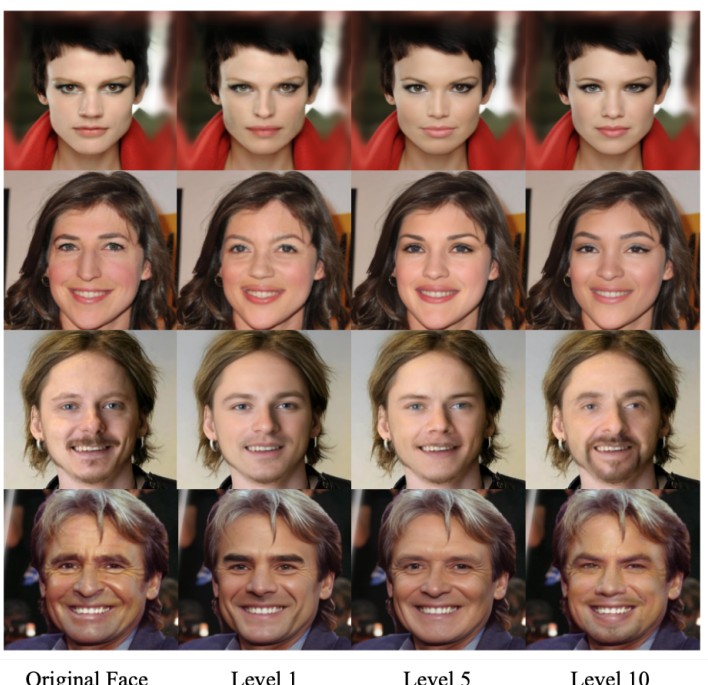

Original Face      Level 1      Level 5      Level 10

Figure 3: Qualitative results of different de-identified level.

We conducted a comparative analysis of our method against state-of-the-art approaches such as DeepPrivacy, CIAGAN, and Repaint. Our model demonstrates superior performance in altering geometry related to identity, particularly facial shapes, while preserving attributes unrelated to identity. We have fully retained elements that are unrelated to identity, such as hair and background, which are crucial for maintaining the naturalness. Furthermore, we report quantitative results in comparison to the aforementioned methods. The results presented in Table 1 indicate that our model outperforms other models in terms of both concealing identity information and preserving attribute information.

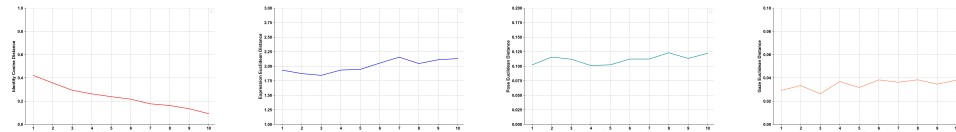

Figure 4: The de-identified performance variation with respect to the de-identified level on the Face-Forensics++. The x-axis indicates the de-identified level and the y-axis indicates different metric values.

|            | PSNR ↑ | EXP ↓ | POSE ↓ | FID ↓ |
|------------|--------|-------|--------|-------|
| DeepPrivacy | 19.7  | 3.36  | 0.2325 | 23.68 |
| CIAGAN      | 18.1  | 3.23  | 0.3024 | 27.78 |
| RePaint     | 21.8  | 4.59  | 0.2012 | 19.67 |
| DiffDeID    | **22.3** | **2.03** | **0.1078** | **12.96** |

Table 1: Quantitative comparison of face de-identification on FaceForensics++ dataset. The up arrow indicates that the larger the value, the better the model performance, and vice versa.

## 4.3 VIDEO APPLICATION ANALYSIS

Fig 5 illustrates the variation of these metrics across frames of the de-identified video. It is evident that the de-identified ID exhibits a small cosine distance value between the source video and the anonymized video, indicating that the identity has been effectively protected. Additionally, the identity consistency cosine distance between frames within the de-identified video is relatively large, signifying that the model maintains a consistent level of de-identified throughout the video stream. Furthermore, the large cosine distance values for other attribute metrics between the source video and the de-identified video indicate that the model has successfully retained the identity-unrelated attributes.

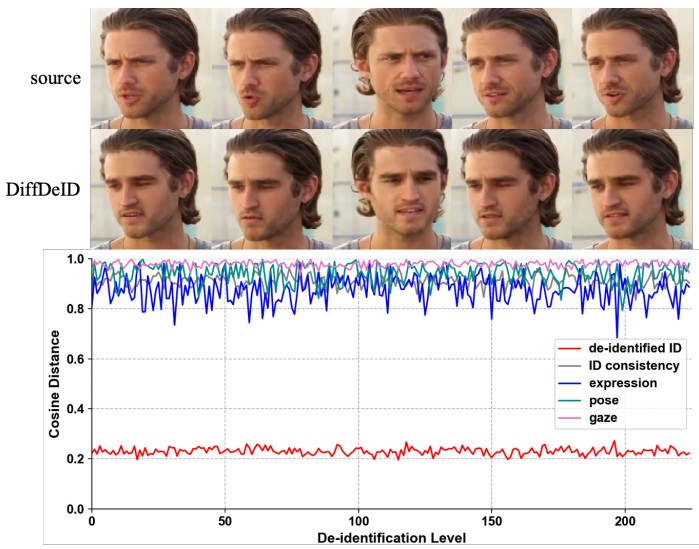

Figure 5: Qualitative and quantitative results of video de-identified.

The quantitative comparative results for video de-identification are presented in the table. Thanks to the explicit facial representation contained within the rendered faces, our method has achieved impressive scores in attribute preservation, which indicates that the model is capable of animating the source facial structure with a high degree of fidelity. The explicit facial representation allows a

more accurate and consistent transfer of face attributes, ensuring that the de-identified output closely mirrors the original video in terms of appearance and expressiveness.

| | EXP ↓ | POSE ↓ | GAZE ↓ | ID ↑ | FID ↓ |
|---|---|---|---|---|---|
| FOMM | 7.01 | 0.0626 | 0.1021 | 0.5622 | 40.21 |
| PIRender | 6.72 | 0.0633 | 0.0977 | 0.5719 | 38.76 |
| NTHS | 7.23 | 0.0795 | 0.1132 | 0.6432 | 38.35 |
| DAM | 7.12 | 0.0618 | 0.0937 | 0.5419 | 43.58 |
| DiffDeID | **5.93** | **0.0424** | **0.0397** | **0.7015** | **36.13** |

Table 2: Quantitative results of video de-identification on VoxCeleb1.

## 5 CONCLUSION

DiffDeID presents a significant advancement in face de-identification by effectively balancing identity protection with the preservation of other facial attributes. Our approach, rooted in diffusion models, employs identity embeddings from expert recognition networks and decoupled 3D identity representations, ensuring both robust identity protection and high utility retention. The novel multi-condition diffusion model and the unified architecture for video facial de-identification further enhance the method's robustness and versatility. Extensive experimentation confirms that DiffDeID surpasses existing methodologies, offering superior natural identity protection and high-quality image details.

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
