# OpenReview forum: "DiffDeID: a Multi-conditional Diffusion-based Method for High Fidelity Face De-indentification with Diversity"
_ICLR.cc/2025/Conference — ICLR 2025 Conference Withdrawn Submission_

### Official Review · Reviewer_5tX1 · 2024-10-19

**Soundness:** 3
**Presentation:** 3
**Contribution:** 2
**Rating:** 5
**Confidence:** 4

**Summary:**

In this paper, the authors propose a diffusion-based de-identification method. Specifically, they aim to replace the identity information in the source images with a synthetic face while preserving original facial attributes such as hair and movement, thereby protecting the privacy of the individuals in the source images. To achieve this, they introduce a multi-condition diffusion model that can sample a person’s identity. Additionally, to facilitate identity swapping, they employ an identity mask to obscure the original facial features.

Overall, the de-identification method proposed in this paper has merits, but several concerns remain. Please refer to the weaknesses.

**Strengths:**

- A diffusion-based de-identification method, which leverages multidimensional identity embeddings, contributes to enhancing user privacy protection.

- Multiple experiments demonstrate the effectiveness of the method.

**Weaknesses:**

- The distinction between de-identification and face swapping: The method proposed in this paper essentially appears to be a form of face swapping, i.e., the well-known deepfake. The key difference is that in deepfake technology, the target face is a real individual, while in this work, the target face is derived from a 3DMM. However, this distinction does not create a clear boundary between the two approaches. The reviewer suggests that the authors should compare their method with state-of-the-art face swapping algorithms.

- Although the experiments demonstrate the high quality of the synthesized faces, there is still a lack of quantitative experiments showing the effectiveness of this method in de-identification. The authors propose a de-identification ID metric, but this metric is not consistently highlighted across all experiments.

- Another concern relates to the practical application. in what scenarios would users need to generate a video with a completely different identity, especially one that is neither animated nor intentionally altered for comedic or entertainment purposes?

- Regarding ethical concerns: Since the proposed method is somewhat related to deepfake technology and involves higher-precision face swapping, it may pose potential risks for harmful applications.

**Questions:**

Please see weaknesses

**Details Of Ethics Concerns:**

- Regarding ethical concerns: Since the proposed method is somewhat related to deepfake technology and involves higher-precision face swapping, it may pose potential risks for harmful applications.

---

### Official Review · Reviewer_uDWy · 2024-10-31

**Soundness:** 3
**Presentation:** 3
**Contribution:** 2
**Rating:** 5
**Confidence:** 4

**Summary:**

The authors propose a multi-conditional Diffusion-based method for face de-identification. The method employs a Latent Diffusion-based ID Sample to generate authentic identity embeddings that are obfuscated from the original identity. Also, a 3D prior is used to improve the robustness.

**Strengths:**

- A Latent Diffusion-based ID Sampler based on DDIM is proposed, which can sample multidimensional identity embeddings with diversity and realism
- DiffDeID pipeline based on a multi-conditional diffusion model for face de-identification is proposed, which can preserve the pose and expression information of the source face image while replacing the identity information

**Weaknesses:**

- For quantitative measures, there is no analysis for the face recognition performance in Table 1, which I think is the most crucial experiment to demonstrate the face de-indentification.
- The visualization is not satisfactory. In Fig. 3, even for level 1, all the faces look different from the original images. Subjective user studies are suggested to consolidate this part.
- No ablation studies to evaluate each proposed component. In particular, I am curious about the difference between with and without 3DMM.

**Questions:**

See the Weakness.

---

### Official Review · Reviewer_CmV1 · 2024-11-03

**Soundness:** 3
**Presentation:** 2
**Contribution:** 3
**Rating:** 6
**Confidence:** 3

**Summary:**

The paper introduces a novel diffusion-based approach, called DiffDeID, suitable for de-identification of face images. The approach relies on four main components, including a facial mask predictor, a 3D Morphable Model (3DMM) for defining facial features, a Latent Diffusion-based ID sampler, used for constructing diverse de-identified identity embeddings, and lastly a multi-conditional diffusion model responsible for producing realistic face images. The de-identification pipeline first entails the extraction of 3D Morphable Model (3DMM) coefficients from an input face image, which include the identity, expression, texture, illumination and pose. In the next step, the identity coefficient is replaced with a de-identified coefficient produced by the Latent Diffusion-based ID sampler. The multi-conditional diffusion model then considers the coefficients and the masked face image to produce replacement face images with a de-identified subject. The proposed solution is trained using the CelebAMask-HQ dataset and its de-identification capabilities are evaluated with the FaceForensics++ dataset. Throughout the experiments the authors showcase that DiffDeID outperforms the state-of-the-art in terms of de-identification while preserving attributes unrelated to identity. The suitability for video de-identification is also explored, by evaluating the proposed solution on the VoxCeleb1 dataset.

**Strengths:**

The paper is written in a clear and concise manner. Differently from existing approaches, DiffDeID utilizes a Latent Diffusion-based ID sampler for sampling new anonymized identity embeddings. When defining 3DMM coefficients the paper also take into account the gaze direction of subjects, which is lacking in the original implementation, despite being a rather crucial feature.  With a new training and inference procedure the proposed architecture also achieves video face de-identification, preserving information not related to the identity. The performed experiments entail both qualitative and quantitative results, exploring the similarity of input and generated identities as well as the preservation of other face coefficients. The approach is compared to three existing solutions where it achieves better results across all measured metrics. Notably, the approach allows for drastically better video face de-identification, which represents a crucial challenge in the real-world application of de-identification methods.

**Weaknesses:**

Despite the listed strengths, the paper suffers from a few major weaknesses, which should be addressed to improve the overall quality of the paper.

1.	The methodology section lacks suitable references to existing works. These are crucial for identifying aspects of the work that are novel and for giving credit to original authors. The experiments section also lacks references to the state-of-the-art that DiffDeID is compared to. Existing approaches and their potential weaknesses should also be explored in more detail in the paper.

2.	Figure 1. should be better annotated to allow the reader to more easily connect parts of the described pipeline with the figure. Additionally, all parts of the pipeline should be depicted, including the input images and the 3DMM model. These changes would allow for better and easier understanding of the proposed approach. The plots and text in Figure 4 are also too difficult to read, due to their small scale. Their readability should be improved by making the figures larger or at least increasing the font size.

3.	The experiments section could be improved introducing the metrics in more detailed manner. For example, peak signal-to-noise ratio (PSNR) is reported in Table 1. but is never mentioned in the paper. It would also be highly beneficial to utilize more metrics to showcase the suitability of the DiffDeID approach. For example, Face Image Quality Assessment (FIQA) measures could reveal valuable insight into the quality of de-identified images [1]. Furthermore, genuine and imposter distributions along with corresponding measures (e.g. Equal Error Rate, Fisher’s Discriminant Ratio), might also provide additional information regarding the separability of real and de-identified identities.

4.	The paper would also benefit from additional ablation studies in the supplementary material for different aspects of the proposed pipeline. For example, the masking of the mouth could be qualitative evaluated. Similarly, the influence of different 3DMM coefficients not related to identity could also be evaluated, e.g. showcasing the images generated without and with control over the different coefficients.

5.	When evaluating video face de-identification it would be beneficial to showcase qualitative samples of state-of-the-art approaches in Figure 5. alongside samples generated by DiffDeID.

**Questions:**

The main points that should be addressed are listed in the weaknesses. Here are a few questions that might spark some discussions:

1.	In the paper you focus on preserving various aspects of faces (e.g. pose), while changing the identity of subjects. Have you perhaps explored how your approach influences other soft biometric features, for example the color of skin? Would it be beneficial to provide such control in the future? Can you provide any comments?

2.	When discussing video de-identification the paper does not mention the time requirements of the method. Can DiffDeID perhaps be ran in real-time, e.g. at a lower level of de-identification? How about other existing approaches? Can you perhaps elaborate more on this topic or even mention this aspect in the paper?

---

### Official Review · Reviewer_6A3g · 2024-11-04

**Soundness:** 2
**Presentation:** 2
**Contribution:** 2
**Rating:** 3
**Confidence:** 4

**Summary:**

This paper presents a face de-identification method that utilizes a multi-condition diffusion model. The proposed method employs the Latent Diffusion-based ID sampler to generate diverse and realistic identity embeddings, which are then used as anonymized identity embeddings for subsequent replacement. Furthermore, a multi-condition diffusion model is employed for facial images to ensure the preservation of image utility.

**Strengths:**

1. The utilization of a Latent Diffusion-based ID Sampler enables the generation of diverse and realistic identity embeddings, serving as anonymized identity embeddings for subsequent substitution.
2. A Multi-Condition Diffusion Model is utilized on facial images to guarantee the maintenance of image utility.

**Weaknesses:**

1. The proposed method neglects the importance of identity recovery. Face de-identification methods should have the ability to recover the original faces when security conditions are satisfied. Otherwise, the distinction between the face de-identification method and face swapping algorithms would be minimal, as face swapping algorithms involve replacing identities while preserving consistency in expressions and poses.
2. The conducted experiments were not sufficiently comprehensive, and there was a lack of ablation experiments to clarify the specific effects of the proposed method.

**Questions:**

1. The proposed method neglects the importance of identity recovery. Face de-identification methods should have the ability to recover the original faces when security conditions are satisfied. Otherwise, the distinction between the face de-identification method and face swapping algorithms would be minimal, as face swapping algorithms involve replacing identities while preserving consistency in expressions and poses.

---

### Official Review · Reviewer_5Cyb · 2024-11-12

**Soundness:** 1
**Presentation:** 2
**Contribution:** 1
**Rating:** 3
**Confidence:** 4

**Summary:**

This work presents a de-identification method call DiffDeid which aims to generate high-fidelity de-identified faces with a balance between anonymity and utility. Diverse identities are generated through sampling on the latent space of a diffusion model.

**Strengths:**

Authors have explored the possibility of performing de-identification tasks with diffusion models.

**Weaknesses:**

-Writing
The motivation of the paper is not good enough. Why we need
-Experimental results
1.Current state-of-the-art diffusion models can generate images with a resolution of 512x512. Authors only show result on images of 256px.
2.The function of several components, which authors claimed to be useful, are not evaluated.  There are no ablation study section.
-Comparison with existing baselines
3.The provided method only compare with Deepprivacy, CIAGAN and Repaint, the newest of which is publish in 2020. Several important methods are missing, e.g. FiT [1], RiDDLE [2].
4.Even in the methods mentioned in the paper, the qualitative results are not shown. Authors should put faces generated from different methods in a single image so reviewers can make a direct judgement
[1] Gu et al Password Conditioned Face id transformer
[2] Li et al. RiDDLE, Reversible and De-diversified De-identification with Latent Encryptor
-Novelty
Except for the latent diffusion models incorporated in the framework,  few novelty is found in the paper. Current SOTA de-id methods can achieve anonymization, which the presented method can not do.
-Figure
I suggest authors to re-write the paper and re-draw some important figures. Many figures in the paper are either too small for me to see the texts(e.g. Fig. 4) or so large that occupies too much space.(e.g. Fig.1)

**Questions:**

Please refer to the weakness part.

---

### Note · Authors · 2024-11-22

I have read and agree with the venue's withdrawal policy on behalf of myself and my co-authors.